# Mortality Related to Air Temperature in European Cities, Based on Threshold Regression Models

**DOI:** 10.3390/ijerph19074017

**Published:** 2022-03-28

**Authors:** Lida Dimitriadou, Panagiotis Nastos, Kostas Eleftheratos, John Kapsomenakis, Christos Zerefos

**Affiliations:** 1Research Centre for Atmospheric Physics and Climatology, Academy of Athens, 10680 Athens, Greece; jkaps@academyofathens.gr (J.K.); zerefos@geol.uoa.gr (C.Z.); 2Laboratory of Climatology and Atmospheric Environment, Department of Geology and Geoenvironment, National and Kapodistrian University of Athens, 15784 Athens, Greece; nastos@geol.uoa.gr (P.N.); kelef@geol.uoa.gr (K.E.); 3Biomedical Research Foundation, Academy of Athens, 11527 Athens, Greece; 4Navarino Environmental Observatory (N.E.O.), 24001 Messinia, Greece; 5Mariolopoulos-Kanaginis Foundation for the Environmental Sciences, 10675 Athens, Greece

**Keywords:** threshold regression analysis, temperature, mortality

## Abstract

There is a wealth of scientific literature that scrutinizes the relationship between mortality and temperature. The aim of this paper is to identify the nexus between temperature and three different causes of mortality (i.e., cardiological, respiratory, and cardiorespiratory) for three countries (Scotland, Spain, and Greece) and eleven cities (i.e., Glasgow, Edinburgh, Aberdeen, Dundee, Madrid, Barcelona, Valencia, Seville, Zaragoza, Attica, and Thessaloniki), emphasizing the differences among these cities and comparing them to gain a deeper understanding of the relationship. To quantify the association between temperature and mortality, temperature thresholds are defined for each city using a robust statistical analysis, namely threshold regression analysis. In a more detailed perspective, the threshold used is called Minimum Mortality Temperature (MMT), the temperature above or below which mortality is at minimum risk. Afterward, these thresholds are compared based on the geographical coordinates of each city. Our findings show that concerning all-causes of mortality under examination, the cities with higher latitude have lower temperature thresholds compared to the cities with lower latitude. The inclusion of the relationship between mortality and temperature in the array of upcoming climate change implications is critical since future climatic scenarios show an overall increase in the ambient temperature.

## 1. Introduction

Much of the emerging scientific literature echoes on the rising concentrations of greenhouse gases resulting in consequences to Earth’s climate. These consequences pose fundamental threats to many sectors, such as human health and mortality, the touristic sector, natural and cultural heritage, the environment, various environmental ecosystems and their biodiversity, etc. The inextricably linked nexus between climate change and human health spans the trajectory of time (past, present, and future) [1]. Climate change is a health emergency and a matter of paramount importance due to its direct as well as indirect consequences on human mortality [2]. Since the far-reaching consequences of climate change are uncharted waters with great uncertainties, the elevation of temperature may result in unpredictable impacts on mortality in the short and long term. In relevant literature, many studies explore the nexus between mortality and temperature (for example, [3,4,5,6]) indicating that mortality increases due to hot and cold climatic conditions. Numerous climatic scenarios scrutinize future climatic changes. Representative Concentration Pathway (also known as RCPs) are the scenarios developed by the Intergovernmental Panel on Climate Change (IPCC) in 2014 [7], universally accepted and applied in numerous scientific fields that are engaged in exploring the consequences of climate change. Worldwide, different countries may be affected in dissimilar ways in the future; however, all people will be exposed to the potential impacts of climate change with higher risk of exposure among frail populations, such as children, pregnant women, the elderly, etc.

Acclimatization to cold environment entails behavioral responses (for instance, wearing additional layers of clothing, use of hats, gloves, scarves, etc.) [8], while adaptation to heat relies on the body’s ability to act as a natural cooling system [9]. Moreover, the body’s susceptibility to heat decreases, thus its adaptability to heat conditions increases [10]. Hence, geographical variations (i.e., diverse climatic conditions) may result in dissimilar consequences on the response of the human body. A point often overlooked is that climate change may be beneficial for some countries characterized by temperate climate, since milder winters may lead to a decrease in human mortality, while in low-latitude countries upcoming hotter and drier conditions may result in mosquito elimination [11]. However, based on future projections, albeit cold-related mortality will decrease, heat-related mortality will exceed the reduction in cold-related mortality [12]. Thus, there is no compensation of the decrease in cold-related mortality, since heat-related mortality is projected to increase more [13].

Mounting evidence shows that countries with various climatic conditions and social characteristics experience climate change in quite diverse ways (for example, [14,15,16,17]). However, it is challenging to compare in a homogeneous way the results of existing studies, which implement a wide range of methodological approaches, alternative measures of temperature, a variety of biometeorological indices, and different causes of mortality across regions with no geographical proximity. As such, this heterogeneity leads to the identification of dissimilar temperature thresholds.

Several authors highlight the relationship between temperature and relationship in both cold and hot conditions (e.g., [13,18,19,20]). Overall, the nexus is identified as a “U-”, “V-”, or “J-” shaped curve [21]. In Europe, differences in mortality are observed during different seasons (i.e., summer and winter) among northern and southern countries [22]. For instance, people that live in countries in North Europe take more efficient protective measures against the cold in contrast to people that live in South Europe [23], since the winter in the south is milder. Additionally, in a study conducted among northern and southern cities (in Finland, Germany, the Netherlands, the U.K., Italy, and Greece), the authors conclude that for all cities under examination, annual cold-related mortality exceeds its respective heat-related mortality [18]. It is critical to emphasize that different countries need diverse preventive measures that capture the needs so as to combat climate change as a variety of different factors burden and affect each population in dissimilar ways. Minimum Mortality Temperature (MMT) is a critical indicator applied to both cold- and heat-related mortality. MMT is defined as the temperature at which mortality is lowest [24,25,26], reflecting human adaptability to temperature in a local scale [25]. MMT varies greatly among different countries [24].

There is a wealth of evidence supporting the overall exceedance of cold-related mortality compared to heat-related mortality. For instance, a study analyzing the nexus among temperature, humidity, and daily mortality for 200 regions in Europe forecasting future scenarios until 2100, concludes that monthly maximum winter mortality exceeds monthly maximum summer mortality [27]. Nonetheless, the authors highlight that in the future heat-related mortality exceeds cold-related mortality and the former compensates the latter [27]. Seasonal mortality in New Zealand reveal excess winter mortality compared to summer mortality due to social characteristics [28]. Moreover, an analysis conducted among the elderly (people over 65 years old) in 2015 in Italy [29], demonstrates that winter mortality exceeds summer mortality. In a more recent study [12], the future effects of climate change on temperature-attributable mortality across 147 regions in 16 European countries are quantified. The study concludes that concerning mortality, cold is ten-fold more harmful than heat [12]. However, in the second half of the 21st century, heat fraction will exceed cold fraction resulting in more deaths due to rising temperatures [12]. In an analysis conducted in a larger scale, the relationship between temperature and mortality was analyzed in 306 communities from 12 countries/regions [30]. The authors highlight that the estimated impacts of temperature on mortality vary among different regions/countries, with cold effects showing delay (lag) and appearing after several days of the initial exposure, while heat-related effects appear immediately [30].

The emerging future effects of climate change on human morbidity and mortality are modeled based on a plethora of various climatic scenarios and regions (see among others [9,19,20,31]). Nonetheless, due to great uncertainty and unknown factors, there is not a single accurate prediction of future mortality. Preventive policy measures focus on national and international scale to combat climate change, along with city and regional scale. Specifically, mitigation action plans shift their emphasis on city scale since the associated risks become more relevant to private and public stakeholders, as well as local administration [14]. The inclusion of the nexus between mortality and temperature in the menu of upcoming climate change implications is critical since future climatic scenarios show an overall increase in the ambient temperature, potentially resulting in augmented mortality. Hence, early warning systems and precautionary policy measures will moderate the effects and ensure preventable mortality.

The aim of the present analysis is to examine the relationship among mortality (cardiological, respiratory, and cardiorespiratory) and temperature for three countries (i.e., Scotland, Spain, and Greece) and their major cities (i.e., Glasgow, Edinburgh, Aberdeen, Dundee, Madrid, Barcelona, Valencia, Seville, Zaragoza, Attica, and Thessaloniki) based on threshold regression analysis to quantify the Minimum Mortality Threshold (MMT), which stands for the lowest mean mortality rate. To our knowledge, this is the first study that combines threshold regression analysis to quantify MMT for eleven different cities, comparing one country in the north and two in the south. Our analysis may provide a pattern for preventive policy measures so as to achieve preventable mortality due to extreme cold and warm conditions.

The article is structured as follows: Section 2, “Materials and Methods”, describes the mean temperature and mortality data for different cities in Scotland, Spain, and Greece, and the relevant methodological framework used. Section 3, “Results”, presents the time series of temperature and mortality data, and the scatter plots between them to reveal the relationship between the two parameters. We then apply the threshold regression analysis between the two parameters to estimate the critical lower and upper temperature values for each city, among which we have the minimum number of deaths. Section 4, “Discussion”, discusses the results that arise from the statistical analysis and finally, Section 5, “Conclusions”, summarizes the main conclusions and message.

## 2. Materials and Methods

### 2.1. Data

The data concern mortality and mean temperature for three countries (Scotland, Spain, and Greece) and eleven cities (i.e., Glasgow, Edinburgh, Aberdeen, Dundee, Madrid, Barcelona, Valencia, Seville, Zaragoza, Attica, and Thessaloniki) on a daily frequency. The data evaluate mortality and not morbidity since mortality data are easier to collect and organize compared to hospital admissions. Moreover, mortality data are binary and thus, easier in which to extract conclusions. Lastly, by taking into account north and south cities, it is easier to compare and draw conclusions for the geographical distribution of different causes of mortality in relation to temperature. All data regarding mean temperature were obtained from NASA Prediction of Worldwide Energy Resources (POWER) (https://power.larc.nasa.gov/data-access-viewer/, accessed on 15 February 2022). The period under examination for Scotland and Spain spans from 1 January 1982 to 31 December 2018 on a daily frequency (13,514 observations). The data concerning mortality in Scotland were received from the National Records of Scotland (https://www.nrscotland.gov.uk/, accessed on 15 February 2022), while the respective data for Spain from the Instituto Nacional de Estadística (https://www.ine.es/, accessed on 15 February 2022). Lastly, the period under examination for the Attica region spans from 1st January 1992 to 31st December 2018 on a daily frequency (9862 observations), whilst for Thessaloniki from 1st January 1999 to 31 December 2016 on a daily frequency (6575 observations). The respective data were obtained from the Hellenic Statistical Authority (https://www.statistics.gr/en/home/, accessed on 15 February 2022). Table 1 includes all the cities, periods, and causes of mortality under examination, starting with the largest cities of each country and in descending order. Where possible, an additional variable called cardiorespiratory mortality was created. Cardiorespiratory mortality is defined as the sum of deaths attributed to cardiological and respiratory diseases. Additionally, to easily compare different cities and regions, mortality is converted to deaths per 100,000 citizens.

### 2.2. Methodological Framework

Several authors have stressed that extreme heat and cold events are inextricably linked to mortality [10,32,33,34]. Since the relationship between mortality and temperature follows a “U-”, “V-”, or “J-”shape curve [21], mortality monotonically increases in low and high temperature and the lowest point of the curve, called Minimum Mortality Threshold (MMT), corresponds to the lowest mean mortality rate [35,36,37]. In essence, MMT is the temperature at which mortality is at minimum risk [38]. MMT is applied for both heat- and cold-related mortality [25]. Hence, it is a helpful indicator to capture the change in the slope for different climatic conditions. In different regions, the slope of the curve (i.e., MMT) is different with higher values revealing larger impact on mortality for a given temperature [35]. The restriction of MMT is that frequently the point that the curve changes its slope is not clearly identified [38]. Moreover, MMT changes throughout the years; thus, a constant MMT (i.e., a constant temperature impact on mortality) is unrealistic [39].

MMT varies for different cities and various causes of mortality and may change throughout the years due to population acclimatization. Since threshold regression models are defined as a set of nonregular regression models that depend on change points or thresholds [40], it is the optimal methodology so as to quantify and interpret MMT and compare the thresholds for the various countries under examination. Additionally, threshold regression analysis is used for nonlinear relationships and as mentioned above, the nexus between mortality and temperature is nonlinear. There is a distinction between threshold regression and change-point analysis, since the former involves modeling nonlinearity and the latter considers time series data primarily concerned with investigating structural changes throughout time [40]. In the present analysis, to quantify MMT, threshold regression analysis was applied, resulting in two threshold temperatures (lower and upper), which are the temperatures below (or above) which the coefficient of mortality changes. All the statistical analysis and the visual representations (figures) were produced using EViews 10 Software.

In the present analysis, discrete threshold regression analysis runs between mortality (cardiological, respiratory, and cardiorespiratory) and mean temperature for 11 cities were estimated, taking into consideration the two threshold breakpoints (lower and upper) [41,42]. Specifically, the formula to calculate the thresholds follows:(1)$$m_{t}=c_{1}\times T\times I_{\left(T<b_{1}\right)}+c_{2}\times T\times I_{\left(b_{1}<T<b_{2}\right)}+c_{3}\times T\times I_{\left(T<b_{3}\right)}+c_{4}+u_{t}$$
where $m_{t}$ is mortality; $c_{1},c_{2},c_{3},c_{4}$ are parameters to be estimated; T is mean temperature; $b_{1},b_{2},b_{3}$ are the temperature thresholds; I(•) is an indicator factor that receives the value one if the condition in the parenthesis is true, while zero otherwise; and $u_{t}$ is the error term.

## 3. Results

In the present analysis, mean temperature is used as a more representative indicator compared to maximum and minimum temperatures, since it captures the average temperature throughout the day [43,44]. Hence, Figure 1 depicts mean temperature for 11 cities on a monthly frequency. As shown, temperature series undergo a distinct seasonal pattern for all cities. To accurately compute and quantify the relationship between mortality and temperature, the seasonality element is removed from both variables using the robust statistical analysis called Seasonal and Trend Decomposition using LOESS (STL Decomposition) that decomposes a time series into three components, namely seasonality, trend, and residual.

As mentioned above, three different causes of mortality are included in the analysis, i.e., cardiological, respiratory, and cardiorespiratory mortality. Cardiorespiratory mortality is the cumulative amount of cardiological and respiratory mortality. Figure 2 presents cardiological mortality for all cities under examination on a monthly basis. Mortality is expressed per 100,000 citizens so as to be easily comparable. Furthermore, the red line represents the trend. As shown, concerning cardiological mortality, by observing the trend line in Scotland, there is a downward trend throughout the years, while for Spain and Greece the trend is either upward or flat. Hence, mortality is either decreased or remains at the same levels.

Figure 3 presents respiratory mortality for all cities under examination on a monthly frequency. Lacking data availability, respiratory mortality for Thessaloniki is not included. Mortality for all cities is expressed per 100,000 citizens in order to be more easily comparable. The red line represents the trend. As shown, concerning respiratory mortality, by observing the trend line in Scotland and Greece there is a flat trend with some spikes, while for Spain the trend is upward for all cities under examination indicating that mortality due to respiratory diseases has increased.

Figure 4 presents cardiorespiratory mortality for all cities under examination. Mortality for all cities is expressed per 100,000 citizens in order to be more easily comparable. The red line represents the trend. As shown, concerning cardiorespiratory mortality, the trend line in Scotland is downward meaning that cardiorespiratory mortality decreases throughout the years, while concerning Spain and Greece the trend is flat with some spikes.

Table 2 presents the cumulative mortality. In essence, Table 2 demonstrates the sum of each cause of death (i.e., cardiological, respiratory, and cardiorespiratory) for all years and all cities under examination. Mortality is expressed per 100,000 citizens so as to be more easily comparable.

Figure 5 depicts scatter plots between cardiological mortality and mean temperature for all the cities under examination. As illustrated, Glasgow, Madrid, Barcelona, Seville, Valencia, and Attica show a U-shaped curve indicating that increased mortality is observed in both low and high temperatures. Furthermore, the U-shaped curves prove the nonlinearity of the nexus between mortality and temperature, meaning that temperature affects mortality in a nonconstant way. This finding is in agreement with the relevant literature that identifies the relationship between mortality and temperature (e.g., [21,38,45]). The lowest point of the curve is the Minimum Mortality Temperature (MMT), namely the temperature with the lowest mortality risk.

Figure 6 depicts scatter plots between respiratory mortality and mean temperature for all the cities under examination. As illustrated, the scatter plots for Madrid and Barcelona have a U-shape.

Figure 7 depicts scatter plots between cardiorespiratory mortality and mean temperature for all the cities under examination. As illustrated, Glasgow, Madrid, Barcelona, Valencia, Seville, and Attica have a distinct U-shaped curve indicating that increased cardiorespiratory mortality is observed for both low and high temperatures. Furthermore, the U-shaped curves prove the nonlinearity of the nexus between mortality and temperature, meaning that temperature affects mortality in a nonconstant way. This finding is in agreement with the relevant literature that identifies the relationship between mortality and temperature (e.g., [21,38,45]). The lowest point of the curve is the Minimum Mortality Temperature (MMT), namely the temperature with the lowest mortality risk.

Table 3 presents the descriptive statistics for each city expressed per 100,000 citizens on a daily frequency for the data set throughout the years. For all cities under examination, the highest cardiological, respiratory, and cardiorespiratory mortality is observed in Seville, Dundee, and Valencia, respectively.

After plotting scatter plots, the nexus between mortality and mean temperature (Figure 5, Figure 6 and Figure 7), discrete threshold regression analysis [41,42] runs, and two temperature thresholds are estimated for every city (i.e., lower and upper). The break dates that the thresholds occur are treated as unknown variables to be estimated [41]. Table 4 jointly presents the thresholds that are estimated for every city under examination, the coordinates of every city (i.e., longitude and latitude), and the statistical significance at the conventional levels of significance (i.e., 1%, 5%, and 10%). The bold numbers show the statistically significant values in all the conventional levels of significance (1%, 5%, and 10%). The levels of significance confirm whether the relationship between mortality and temperature produce a U- or J-shaped curve. If the temperature thresholds are not statistically significant, there is no change in the slope of the curve. By comparing north and south countries, it is shown that temperature thresholds are lower in the former case and higher in the latter for different causes of mortality. Additionally, concerning Scotland, some temperature thresholds are not statistically significant.

The temperature thresholds are presented in Table 4. The lower and upper thresholds of each cause of mortality are compared with the respective thresholds for every city/region under examination. The figures below show the geographical distribution of the calculated temperature thresholds. The cities whose thresholds are not statistically significant thresholds are not included in the figures. In Figure 8, lower and upper temperature thresholds are displayed by different longitude and latitude concerning cardiological mortality. Different colors represent different countries (i.e., green represents Scotland, red represents Spain, and blue represents Greece). The interpretation of the figures concerning cardiological mortality is that concerning latitude, cities located in the north have lower temperature thresholds (both lower and upper) than cities located in the south. Hence, latitude is inversely related to temperature: the higher the latitude, the lower the temperature threshold, and vice versa. Concerning longitude, the thresholds fluctuate in both low and high longitude.

Figure 9 presents the geographical distribution of respiratory mortality. Different colors represent different countries (i.e., green represents Scotland, red represents Spain, and blue represents Greece). The cities whose thresholds are not statistically significant thresholds are not included in the figures. The interpretation of the figures concerning respiratory mortality is that concerning latitude, cities located in the north have lower temperature thresholds (both lower and upper) than cities located in the south. Hence, latitude is inversely related to temperature: the higher the latitude, the lower the temperature threshold, and vice versa. Concerning longitude, the thresholds fluctuate in both low and high longitude.

Figure 10 presents the geographical distribution of cardiorespiratory mortality. Different colors represent different countries (i.e., green represents Scotland, red represents Spain, and blue represents Greece). The cities whose thresholds are not statistically significant thresholds are not included in the figures. The interpretation of the figures concerning cardiorespiratory mortality is that concerning latitude, cities located in the north have lower temperature thresholds (both lower and upper) than cities located in the south. Hence, latitude is inversely related to temperature: the higher the latitude the lower the temperature threshold, and vice versa. Concerning longitude, the thresholds fluctuate in both low and high longitude.

## 4. Discussion

The aim of the present study is to examine the nexus between mean temperature (displayed in Figure 1) and three causes of mortality (i.e., cardiological, respiratory, and cardiorespiratory) (presented analytically in Figure 2, Figure 3 and Figure 4) for three countries (one north (i.e., Scotland) and two south countries (Spain and Greece)) and eleven major cities (i.e., Glasgow, Edinburgh, Aberdeen, Dundee, Madrid, Barcelona, Valencia, Seville, Zaragoza, Attica, and Thessaloniki). Afterward, the interrelationship between mortality and temperature is shown in scatter plots (Figure 5, Figure 6 and Figure 7). Specifically, by running threshold regression analysis, lower and upper temperature thresholds (i.e., the minimum mortality temperature) are quantified concerning cardiological, respiratory, and cardiorespiratory mortality. The data concern mortality and not morbidity since mortality data are easier to collect and organize compared to hospital admissions. Moreover, mortality data are binary and easier from which to extract conclusions. Lastly, by taking into account north and south cities, it is easier to compare and draw conclusions for the geographical distribution of different causes of mortality in relation to temperature.

The results show that concerning all causes of mortality (i.e., cardiological, respiratory, and cardiorespiratory), the cities located in the north have lower temperature thresholds (both lower and upper) than cities located in the south (i.e., the greater the latitude, the lower the threshold, and vice versa), while concerning longitude, the thresholds fluctuate in both low and high longitude. Since climate change is forecasted to elevate ambient temperature, resulting in excess mortality [13], it is critical to quantify in advance the existing nexus between mortality and temperature in various climatic conditions so as to be able to predict and prevent future mortality. Additionally, by estimating and comparing the mortality burden due to temperature in different countries, conclusions can be extracted in order for policy measures to be applied. By comparing north and south countries, cities can be compared due to the normalization applied (mortality expressed per 100,000 citizens) and conclusions can be extracted based on the longitude and the latitude of a city. This information is useful since (i) population adaptation based on different climatic scenarios is easier to compare; (ii) from a sociological perspective, lower temperatures are linked to behavioral responses (using additionally layers of clothing, gloves, hats, etc.); and (iii) if the temperature increases due to climate change, there is a chance that a shift of temperature thresholds will occur, thus creating a milder environment in northern countries and a more tropical environment to southern countries.

While the relationship between temperature and mortality is thoroughly examined, the novelty of the present paper lies in the unique data set used that spans a large period of time on a daily frequency for all the cities under examination, and the comparison of eleven cities with different climatic conditions for various causes of mortality. Furthermore, the application of discrete threshold regression analysis on 11 cities with diverse climatic characteristics addressed all of the abovementioned issues. In the relevant literature, numerous studies quantify MMT thresholds. For instance, a study conducted in 2002 [46] examined the nexus between temperature and relationship between eleven cities in the eastern United States during 1973–1994. The authors associate MMT with relative risk [46]. Similarly, concerning Oslo (Norway) from 1990 to 1995, MMT was associated with relative risk [47]. In a more recent study [48], MMT was quantified for people over 65 years old for the Netherlands, concluding that MMT increases throughout the years. As mentioned above, MMT varies greatly for different cities and countries. Specifically, MMT ranges from a daily mean temperature between 10 and 12 °C in Oslo (Norway) [47] to 27 °C in Miami (FL, USA) [46]. Both studies accounted for cardiovascular, respiratory, and all-causes mortality. However, a direct comparison of these studies may not be accurate due to the heterogeneity of the countries, time series, periods, and methodological frameworks used.

## 5. Conclusions

Since climate change is a modern-day health threat, the inextricable link between mortality and temperature should be examined so as to provide preventive measures in time. The aim of the present study was to investigate the nexus between temperature and three causes of mortality (i.e., cardiological, respiratory, and cardiorespiratory) for three countries (one north country (i.e., Scotland) and two south countries (Spain and Greece)) and eleven major cities (i.e., Glasgow, Edinburgh, Aberdeen, Dundee, Madrid, Barcelona, Valencia, Seville, Zaragoza, Attica, and Thessaloniki). The comparison of different cities with different social and climatic characteristics provided a broader perspective as to how each city will be affected due to temperature rise in the foreseeable future. In conclusion, we find that northern cities (Glasgow, Edinburgh, Aberdeen, and Dundee) have lower temperature thresholds than southern cities (Madrid, Barcelona, Valencia, Seville, Zaragoza, Athens, and Thessaloniki), which explains better adaptation of the northern cities to cold conditions, while the southern cities have better adaptation to heat compared to northern cities. Turning our focus to mortality, it is critical to observe that all temperature thresholds concerning southern cities are statistically significant for all causes of mortality under investigation, and that cardiological and cardiorespiratory mortality have higher lower and upper temperature thresholds indicating that temperature affects cardiological and cardiorespiratory mortality in a statistically significant way. As temperature continues to rise, its deleterious consequences impact all aspects of everyday life. To that end, further research concerning future climatic conditions and their effects on human mortality is necessary. Hence, early warning systems together with precautionary policy measures will moderate the effects and ensure preventable mortality.

Based on a European perspective, the main message from the present study follows: (i) Different climatic conditions affect different populations and since climate change will affect mortality globally but not uniformly; it is critical to know how different regions are affected. (ii) As the present analysis adds to the relevant literature, temperature and mortality are indeed linked and projections of future rise of ambient temperature will affect mortality. Lastly, (iii) early warning systems are important tools so as to prevent future mortality due to elevated temperature. Hence, further research concerning implementation of early warning systems is needed.

## Figures and Tables

**Figure 1 ijerph-19-04017-f001:**
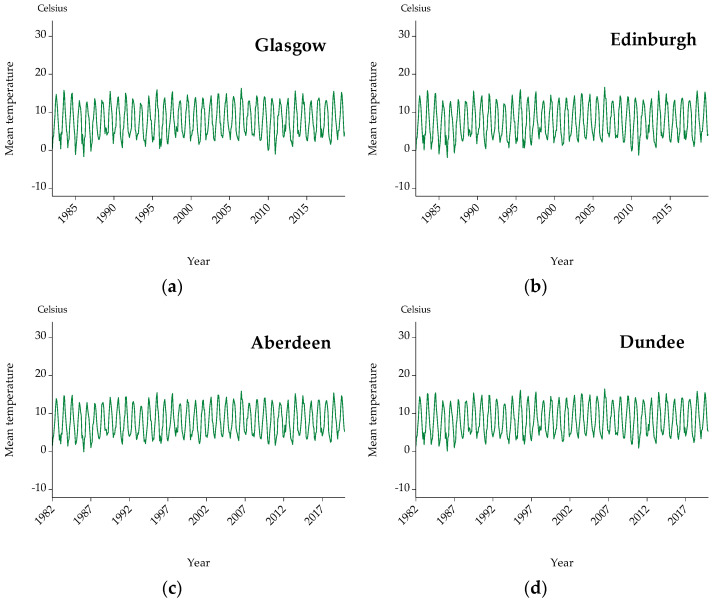
Graphs for mean temperature on a monthly frequency: (**a**) Glasgow, (**b**) Edinburgh, (**c**) Aberdeen, (**d**) Dundee, (**e**) Madrid, (**f**) Barcelona, (**g**) Valencia, (**h**) Seville, (**i**) Zaragoza, (**j**) Athens, and (**k**) Thessaloniki.

**Figure 2 ijerph-19-04017-f002:**
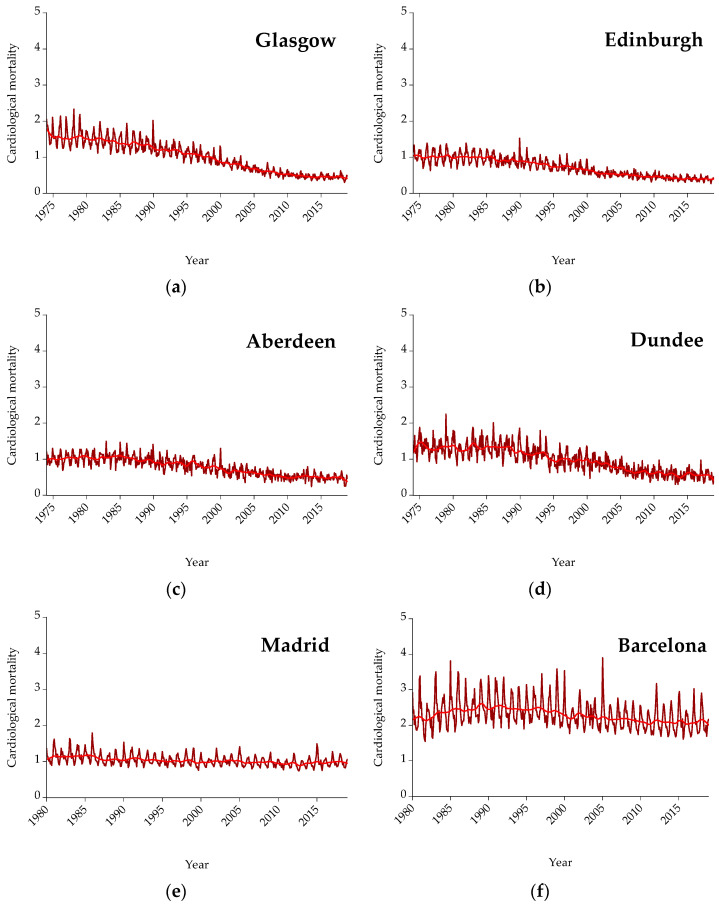
Graphs for cardiological mortality on a monthly frequency: (**a**) Glasgow, (**b**) Edinburgh, (**c**) Aberdeen, (**d**) Dundee, (**e**) Madrid, (**f**) Barcelona, (**g**) Valencia, (**h**) Seville, (**i**) Zaragoza, (**j**) Athens, and (**k**) Thessaloniki. Mortality is expressed per 100,000 citizens.

**Figure 3 ijerph-19-04017-f003:**
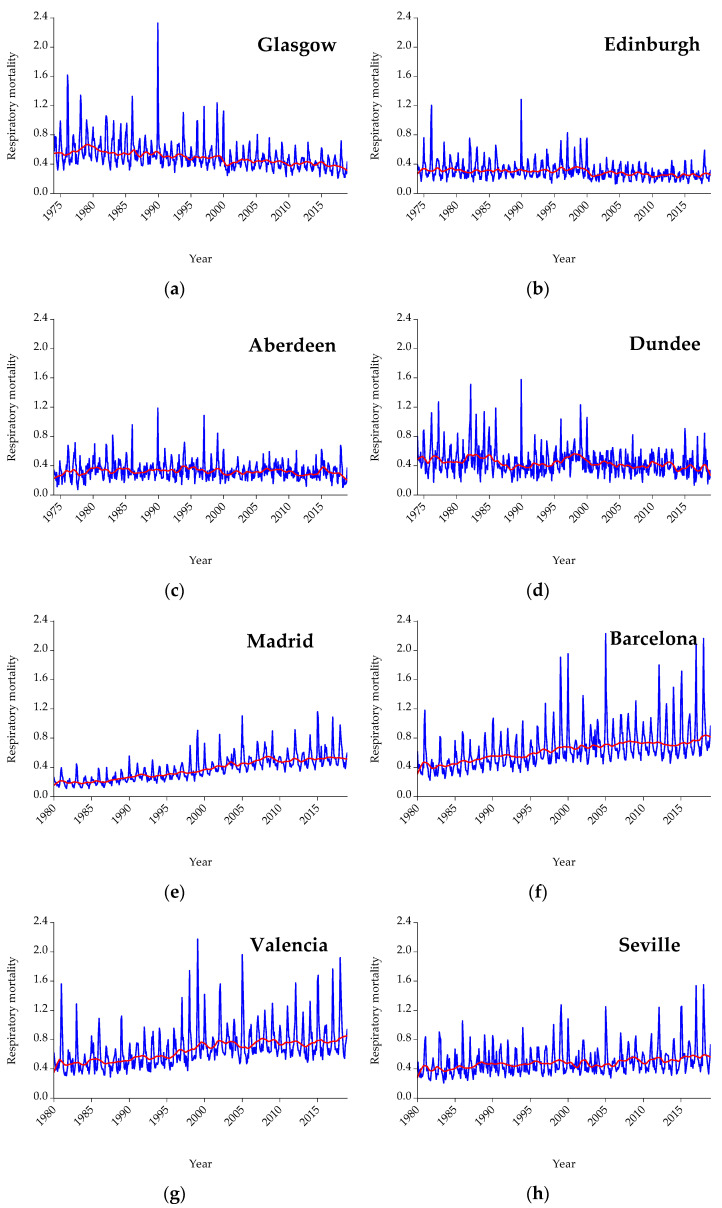
Graphs for respiratory mortality on a monthly frequency: (**a**) Glasgow, (**b**) Edinburgh, (**c**) Aberdeen, (**d**) Dundee, (**e**) Madrid, (**f**) Barcelona, (**g**) Valencia, (**h**) Seville, (**i**) Zaragoza, and (**j**) Athens. Mortality is expressed per 100,000 citizens.

**Figure 4 ijerph-19-04017-f004:**
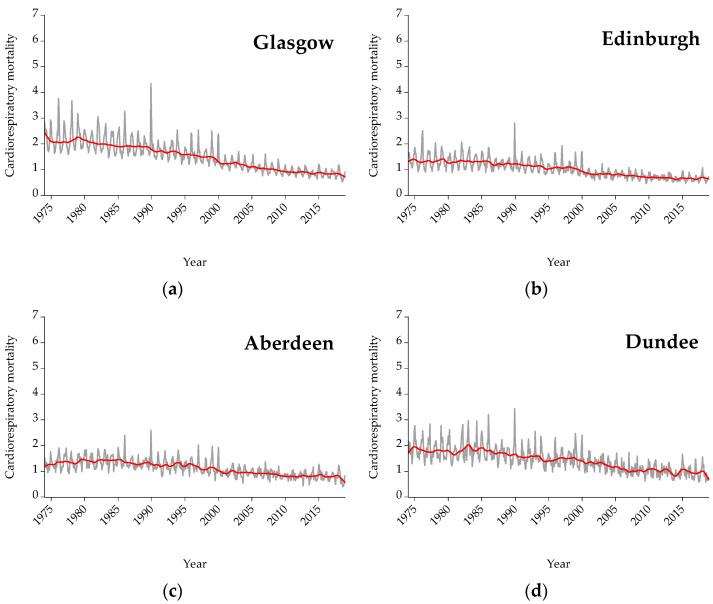
Graphs for cardiorespiratory mortality on a monthly frequency: (**a**) Glasgow, (**b**) Edinburgh, (**c**) Aberdeen, (**d**) Dundee, (**e**) Madrid, (**f**) Barcelona, (**g**) Valencia, (**h**) Seville, (**i**) Zaragoza, and (**j**) Athens. Mortality is expressed per 100,000 citizens.

**Figure 5 ijerph-19-04017-f005:**
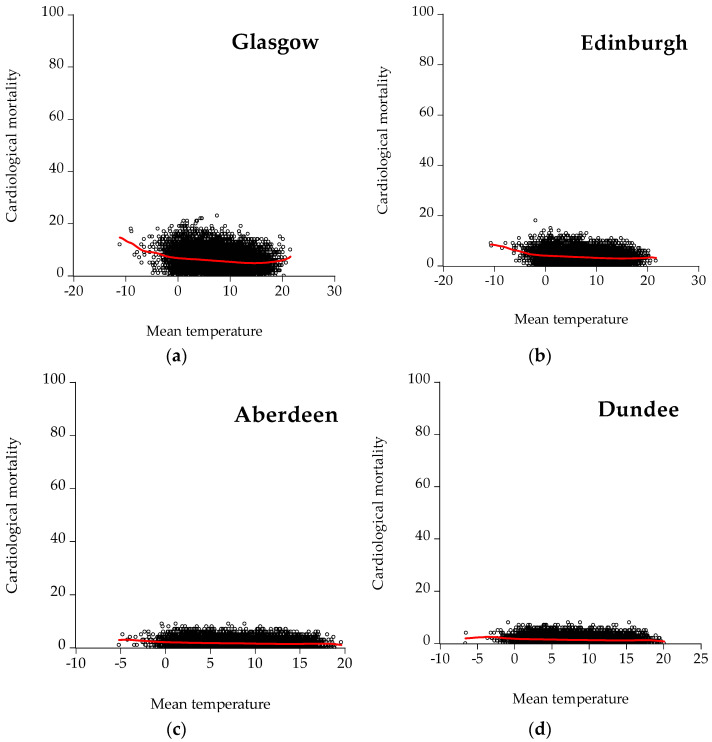
Scatter plots for cardiological mortality: (**a**) Glasgow, (**b**) Edinburgh, (**c**) Aberdeen, (**d**) Dundee, (**e**) Madrid, (**f**) Barcelona, (**g**) Valencia, (**h**) Seville, (**i**) Zaragoza, (**j**) Athens, and (**k**) Thessaloniki. The red line represents the kernel fit line.

**Figure 6 ijerph-19-04017-f006:**
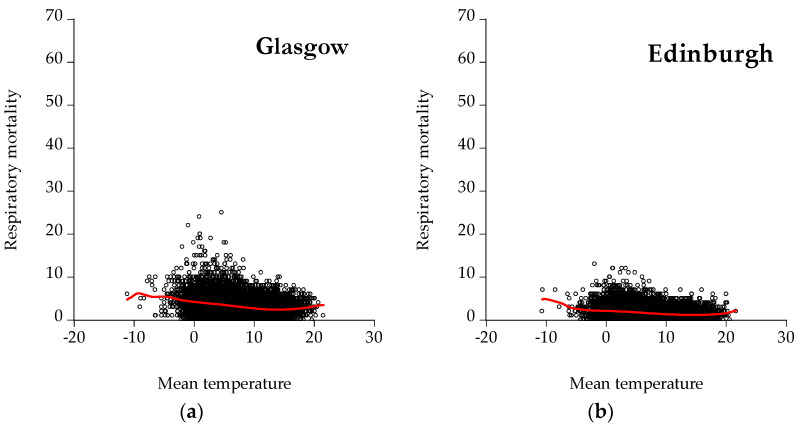
Scatter plots for respiratory mortality: (**a**) Glasgow, (**b**) Edinburgh, (**c**) Aberdeen, (**d**) Dundee, (**e**) Madrid, (**f**) Barcelona, (**g**) Valencia, (**h**) Seville, (**i**) Zaragoza, and (**j**) Athens. The red line represents the kernel fit line.

**Figure 7 ijerph-19-04017-f007:**
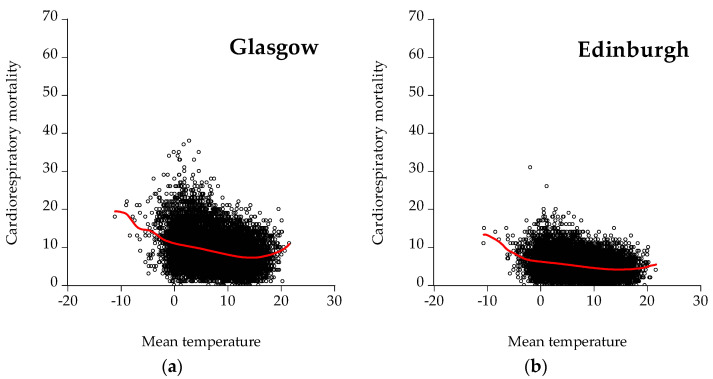
Scatter plots for cardiorespiratory mortality: (**a**) Glasgow, (**b**) Edinburgh, (**c**) Aberdeen, (**d**) Dundee, (**e**) Madrid, (**f**) Barcelona, (**g**) Valencia, (**h**) Seville, (**i**) Zaragoza, and (**j**) Athens. The red line represents the kernel fit line.

**Figure 8 ijerph-19-04017-f008:**
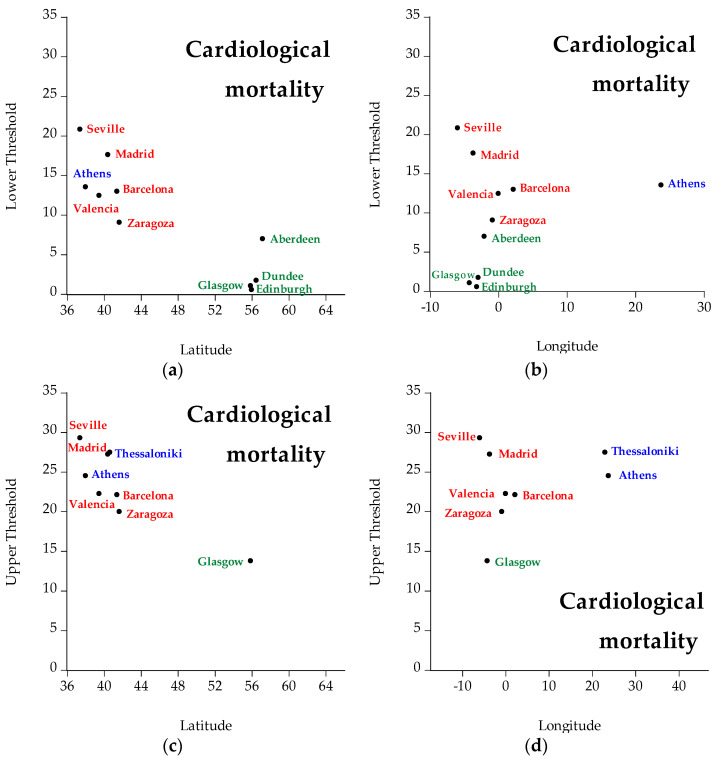
Geographical distribution (i.e., latitude, longitude) of cardiological mortality of all regions under examination, specifically: (**a**) represents lower temperature thresholds for cardiological mortality taking into account latitude, (**b**) represents lower temperature thresholds for cardiological mortality taking into account longitude, (**c**) represents upper temperature thresholds for cardiological mortality taking into account latitude, and (**d**) represents upper temperature thresholds for cardiological mortality taking into account longitude. Green represents Scotland, red represents Spain, and blue represents Greece.

**Figure 9 ijerph-19-04017-f009:**
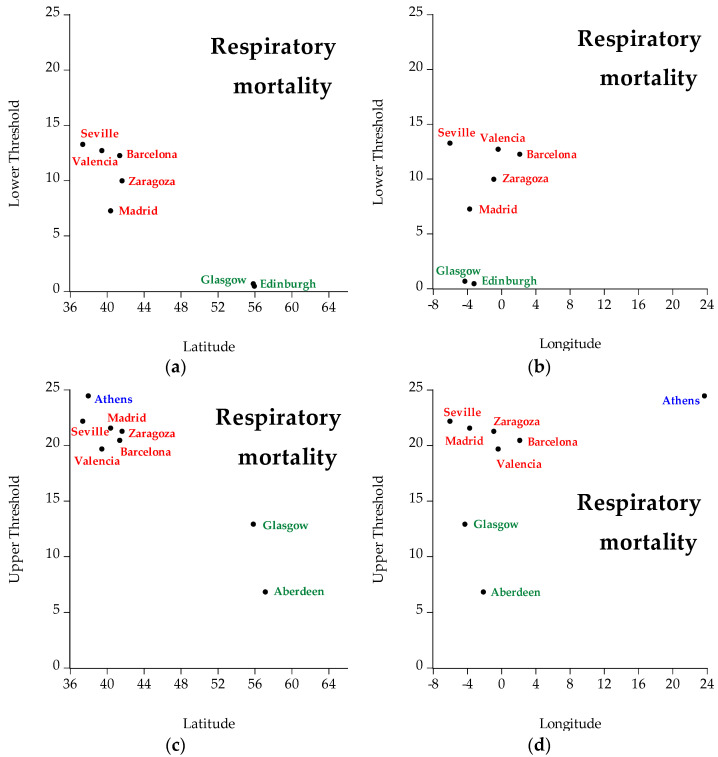
Geographical distribution (i.e., latitude, longitude) of respiratory mortality of all regions under examination, specifically: (**a**) represents lower temperature thresholds for respiratory mortality taking into account latitude, (**b**) represents lower temperature thresholds for respiratory mortality taking into account longitude, (**c**) represents upper temperature thresholds for respiratory mortality taking into account latitude, and (**d**) represents upper temperature thresholds for respiratory mortality taking into account longitude. Green represents Scotland, red represents Spain, and blue represents Greece.

**Figure 10 ijerph-19-04017-f010:**
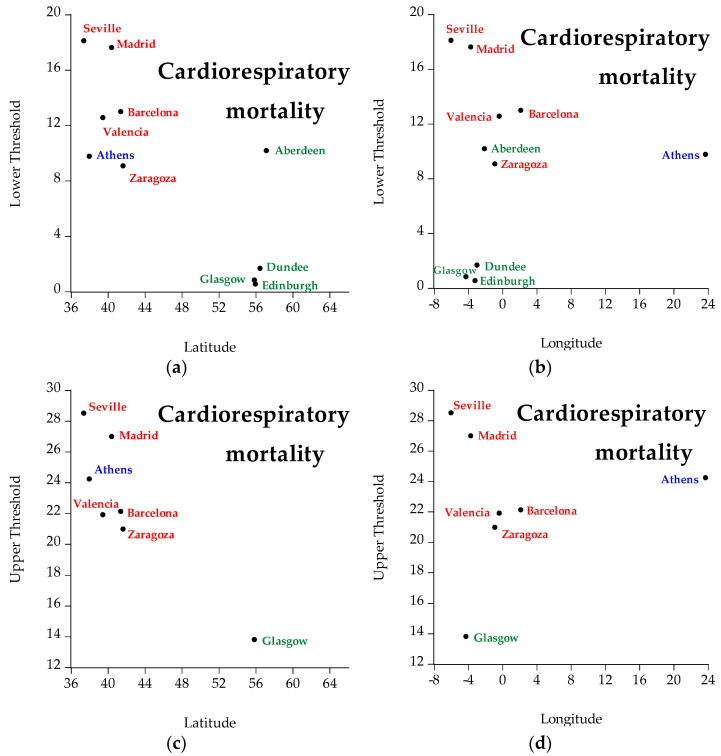
Geographical distribution (i.e., latitude, longitude) of cardiorespiratory mortality of all regions under examination, specifically: (**a**) represents lower temperature thresholds for cardiorespiratory mortality taking into account latitude, (**b**) represents lower temperature thresholds for cardiorespiratory mortality taking into account longitude, (**c**) represents upper temperature thresholds for cardiorespiratory mortality taking into account latitude, and (**d**) represents upper temperature thresholds for cardiorespiratory mortality taking into account longitude. Green represents Scotland, red represents Spain, and blue represents Greece.

**Table 1 ijerph-19-04017-t001:** List of cities and regions under examination.

Country	Region	Period	Cause of Mortality
Scotland	Glasgow	1974–2018	Cardiological, respiratory, cardiorespiratory
	Edinburgh	1974–2018	Cardiological, respiratory, cardiorespiratory
	Aberdeen	1974–2018	Cardiological, respiratory, cardiorespiratory
	Dundee	1974–2018	Cardiological, respiratory, cardiorespiratory
Spain	Madrid	1980–2018	Cardiological, respiratory, cardiorespiratory
	Barcelona	1980–2018	Cardiological, respiratory, cardiorespiratory
	Valencia	1980–2018	Cardiological, respiratory, cardiorespiratory
	Seville	1980–2018	Cardiological, respiratory, cardiorespiratory
	Zaragoza	1980–2018	Cardiological, respiratory, cardiorespiratory
Greece	Attica	1992–2016	Cardiological, respiratory, cardiorespiratory
	Thessaloniki	1999–2016	Cardiological

**Table 2 ijerph-19-04017-t002:** Cumulative amount of mortality (per 100,000 citizens).

City	Cardiological Mort.	Respiratory Mort.	Cardiorespiratory Mort.
Glasgow	16,535	8464	24,999
Edinburgh	13,871	5858	19,729
Aberdeen	13,762	5800	19,562
Dundee	15,579	7193	22,772
Madrid	9629	3392	13,021
Barcelona	12,130	3362	15,492
Valencia	38,125	10,281	48,406
Seville	30,987	7237	38,224
Zaragoza	18,193	5425	23,618
Attica	8840	2796	11,636
Thessaloniki	5903	-	-
Total	183,554	59,808	243,362

**Table 3 ijerph-19-04017-t003:** Descriptive statistics of mortality (per 100,000 citizens).

	Min	Mean	Max	St. Deviation
	Glasgow			
Cardiological mortality	0.00	1.02	3.95	0.0
Respiratory mortality	0.00	0.51	3.95	0.36
CR ^a^	0.00	1.54	6.00	0.78
	Edinburgh			
Cardiological mortality	0.00	0.72	3.43	0.45
Respiratory mortality	0.00	0.31	2.86	0.28
CR ^a^	0.00	1.03	5.90	0.56
	Aberdeen			
Cardiological mortality	0.00	0.79	4.37	0.63
Respiratory mortality	0.00	0.33	3.94	0.40
CR ^a^	0.00	1.12	5.69	0.76
	Dundee			
Cardiological mortality	0.00	0.99	6.03	0.88
Respiratory mortality	0.00	0.45	4.02	0.58
CR ^a^	0.00	1.45	7.37	1.08
	Madrid			
Cardiological mortality	0.15	1.03	2.40	0.25
Respiratory mortality	0.00	0.38	1.60	0.21
CR ^a^	0.15	1.41	3.35	0.34
	Barcelona			
Cardiological mortality	0.68	2.31	5.43	0.57
Respiratory mortality	0.00	0.66	3.15	0.36
CR ^a^	0.74	2.97	7.96	0.78
	Valencia			
Cardiological mortality	0.37	2.53	6.26	0.75
Respiratory mortality	0.00	0.68	3.31	0.40
CR ^a^	0.61	3.21	9.58	0.93
	Seville			
Cardiological mortality	0.00	2.13	7.39	0.73
Respiratory mortality	0.00	0.50	2.99	0.33
CR ^a^	0.00	2.63	8.53	0.90
	Zaragoza			
Cardiological mortality	0.00	1.20	3.41	0.46
Respiratory mortality	0.00	0.36	2.52	0.27
CR ^a^	0.15	1.56	5.04	0.57
	Athens			
Cardiological mortality	0.21	0.74	2.32	0.19
Respiratory mortality	0.00	0.23	0.94	0.12
CR ^a^	0.26	0.97	2.69	0.25
	Thessaloniki			
Cardiological mortality	0.00	0.38	1.12	0.15

^a^ CR = cardiorespiratory mortality indicates the sum of cardiological and respiratory mortality.

**Table 4 ijerph-19-04017-t004:** Temperature thresholds for different countries.

	Country	Region/Cities	Longitude	Latitude	Lower Threshold	Upper Threshold
CM	Scotland	Glasgow	−4.25143	55.86092	**1.04**	**13.77**
		Edinburgh	−3.18827	55.95325	**0.55**	14.53
		Aberdeen	−2.09908	57.14965	**6.98**	14.39
		Dundee	−2.97070	56.46200	**1.73**	15.20
RM	Scotland	Glasgow	−4.25143	55.86092	**0.65**	**12.9** **0**
		Edinburgh	−3.18827	55.95325	**0.43**	11.06
		Aberdeen	−2.09908	57.14965	4.37	**6.81**
		Dundee	−2.97070	56.46200	3.76	9.46
CRM	Scotland	Glasgow	−4.25143	55.86092	**0.82**	**13.8** **0**
		Edinburgh	−3.18827	55.95325	**0.53**	11.29
		Aberdeen	−2.09908	57.14965	**10.17**	14.39
		Dundee	−2.97070	56.46200	**1.67**	15.20
CM	Spain	Madrid	−3.70379	40.41678	**17.61**	**27.24**
		Barcelona	2.15401	41.39021	**12.98**	**22.13**
		Valencia	−0.375000	39.46667	**12.45**	**22.26**
		Seville	−5.99407	37.39253	**20.83**	**29.31**
		Zaragoza	−0.88771	41.64969	**9.07**	**19.98**
RM	Spain	Madrid	−3.70379	40.41678	**7.24**	**21.54**
		Barcelona	2.15401	41.39021	**12.24**	**20.44**
		Valencia	−0.375000	39.46667	**12.69**	**19.65**
		Seville	−5.99407	37.39253	**13.25**	**22.16**
		Zaragoza	−0.88771	41.64969	**9.95**	**21.25**
CRM	Spain	Madrid	−3.70379	40.41678	**17.61**	**26.98**
		Barcelona	2.15401	41.39021	**12.98**	**22.12**
		Valencia	−0.375000	39.46667	**12.55**	**21.90**
		Seville	−5.99407	37.39253	**18.10**	**28.50**
		Zaragoza	−0.88771	41.64969	**9.07**	**20.97**
CM	Greece	Athens	23.72754	37.98381	**13.55**	**24.53**
		Thessaloniki	22.94741	40.62927	**7.61**	**27.48**
RM	Greece	Athens	23.72754	37.98381	**15.00**	**24.44**
CRM	Greece	Athens	23.72754	37.98381	**9.76**	**24.23**

Notes: CM indicates cardiological mortality, RM indicates respiratory mortality, and CRM indicates cardiorespiratory mortality. Bold numbers are statistically significant at confidence interval ≥90%.

## Data Availability

Not applicable.

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
