# Peer review of "Mortality Related to Air Temperature in European Cities, Based on Threshold Regression Models"

_ijerph, 2022, doi:10.3390/ijerph19074017_

Round 1

Reviewer 1 Report

1. This paper does not propose the causal relationship between temperature and diseases such as cardiology, respiratory and cardiorespiratory, nor can it define the degree of disease caused by temperature in the human body. The structure of the article is too arbitrary.

2.Fig.1 to Fig.3 can not understand the data changes, please adjust the drawing surface.

3.This article should add the descriptions of Fig.1 to Fig.3 to improve the current deficiency of only 5 lines of text, so that readers can better understand the connotation of the data.

4.Line 217 should detail the operation of the threshold regression model and the analysis results for better readability.

Author Response

Dear Reviewer,

We would to thank for your fruitful comments which have greatly helped us to improve our manuscript. Please find our answers to your comments in the attached file.

With kind regards,

Kostas Eleftheratos

Reviewer 2 Report

The paper "Mortality related to air temperature in European cities, based on Threshold Regression models" by Lida Dimitriadou, Panagiotis Nastos, Kostas Eleftheratos, John Kapsomenakis and Christos Zerefos analyze the relationship between temperature and three different causes of mortality for three countries and eleven cities in Europe. To quantify the association between temperature and mortality, temperature thresholds are defined for each city using the Threshold Regression Analysis. It is concluded that the cities with higher latitude have lower temperature thresholds compared to the cities with lower latitude.

The subject of the paper is interesting and the methods which are used to examine mortality related to air temperature are correct and adequate. However, the authors can explain MMT in more details.

In my opinion, the paper requires minor revision before it can be published in International Journal of Environmental Research and Public Health.

Specific comments:

 Please, add a Figure 1 with countries, cities and associated values of max T, min T and mean T.

Page 4, Methodological framework: explain Minimum Mortality Threshold (MMT).

Explain Table 2

Line 231 Correct English.

Lines 278-289 Rephrase in one sentence.

Author Response

(The authors gave the same response as above.)

Reviewer 3 Report

I read the paper with interest. A very important aspect is analyzed in it. In terms of language and grammar, the work is written carefully. I only recommend a minor spell check. The language is even poetic in some places. In general, I consider your work to be necessary and written quite well. I recommend major revision because some corrections may be time-consuming. I think that after taking them into account, the work will be ready for publication. My main remarks relate to the introduction, where I would like you to introduce a little more broadly the context of the influence of temperature on mortality - also its indirect influence. The methods should be supplemented with a description of the method you are using. I also recommend a slightly broader discussion at the end, due to the importance of the topic, so that your conclusions are clearly formulated and possibly linked causes with results.

Lines 48-75

In these paragraphs, you generally discuss the effects of temperature and climate on health and mortality. Please add in the introduction that temperature influences mortality not only directly but also indirectly. It has been proved in a paper published in 2021 in the special issue of SENSORS MDPI Sensors for Air Quality Monitoring that temperature is the main factor for PM pollution generation. This is an interesting example because in Central Europe there is only one country whose smog problem is so great that in world rankings its cities are higher than the most polluted developing cities in Asia. It is estimated that there are as many as 5% of excess deaths due to too much exposure to PM2.5 and PM10 and this is related to temperature. Temperature is the main trigger for the production of these pollutants. I believe that despite the fact that you do not study this city directly, this information as an interesting exception on a European scale should be included in the introduction.

Lines 153 - 175 

The methodological framework must be extended. Please elaborate more about Threshold Regression models. Providing just a reference is not sufficient. Please show which equation was used as threshold model etc. and general flow. Why this methodology is good for this study and what are its pitfalls?

Line 183-184 - please add what is the uncertainty resulting from this assumption

Figure 1-3 nothing was mentioned about them in the discussion. I understand that they are shown as more input data, but I would like the Authors to refer to them in a few sentences and to what can be seen on them and why

Figures 4-6 please add a legend to the colors used on the graphs. 

Discussion

After reading the discussion, I feel some kind of scientific hunger. In some places, I feel more like reading a report. I would like the Authors to elaborate more and try to answer the question "why this is so" in a slightly broader perspective. These are very interesting data and analyzes important from the European point of view, and your message should be clear.

Author Response

(The authors gave the same response as above.)

Round 2

Reviewer 3 Report

In my opinion, the paper in this version is ready for publication. I have no more comments. 

Greetings to the authors